# Constant-Expansion Suffices for Compressed Sensing with Generative Priors

**Constantinos Daskalakis**
MIT
costis@mit.edu

**Dhruv Rohatgi**
MIT
drohatgi@mit.edu

**Manolis Zampetakis**
MIT
mzampet@mit.edu

## Abstract

Generative neural networks have been empirically found very promising in providing effective structural priors for compressed sensing, since they can be trained to span low-dimensional data manifolds in high-dimensional signal spaces. Despite the non-convexity of the resulting optimization problem, it has also been shown theoretically that, for neural networks with random Gaussian weights, a signal in the range of the network can be efficiently, approximately recovered from a few noisy measurements. However, a major bottleneck of these theoretical guarantees is a network *expansivity* condition: that each layer of the neural network must be larger than the previous by a logarithmic factor. Our main contribution is to break this strong expansivity assumption, showing that *constant* expansivity suffices to get efficient recovery algorithms, besides it also being information-theoretically necessary. To overcome the theoretical bottleneck in existing approaches we prove a novel uniform concentration theorem for random functions that might not be Lipschitz but satisfy a relaxed notion which we call "pseudo-Lipschitzness." Using this theorem we can show that a matrix concentration inequality known as the *Weight Distribution Condition (WDC)*, which was previously only known to hold for Gaussian matrices with logarithmic aspect ratio, in fact holds for constant aspect ratios too. Since WDC is a fundamental matrix concentration inequality in the heart of all existing theoretical guarantees on this problem, our tighter bound immediately yields improvements in all known results in the literature on compressed sensing with deep generative priors, including one-bit recovery, phase retrieval, and more.

## 1 Introduction

Compressed sensing is the study of recovering a high-dimensional signal from as few measurements as possible, under some structural assumption about the signal that pins it into a low-dimensional subset of the signal space. The assumption that has driven the most research is sparsity; it is well known that a $k$-sparse signal from $\mathbb{R}^n$ can be efficiently recovered from only $O(k \log n)$ linear measurements [4]. Numerous variants of this problem have been studied, e.g. tolerating measurement noise, recovering signals that are only approximately sparse, recovering signals from phaseless measurements or one-bit measurements, to name just a few [1].

However, in many applications sparsity in some basis may not be the most natural structural assumption to make for the signal to be reconstructed. Given recent strides in performance of generative neural networks [6, 5, 11, 3, 12], there is strong evidence that data from some domain $\mathcal{D}$, e.g. faces, can be used to identify a deep neural network $G : \mathbb{R}^k \to \mathbb{R}^n$, where $k \ll n$, whose range $G(x)$ over varying "latent codes" $x \in \mathbb{R}^k$ covers well the objects of $\mathcal{D}$. Thus, if we want to perform compressed sensing on signals from this domain $\mathcal{D}$, the machine learning paradigm suggests that a reasonable structural assumption to make is that the signal lies in the range of $G$, suggesting the following problem, first proposed in [2]:

> COMPRESSED SENSING WITH A DEEP GENERATIVE PRIOR (CS-DGP)
> **Given:** Deep neural network $G : \mathbb{R}^k \to \mathbb{R}^n$; measurement matrix $A \in \mathbb{R}^{m \times n}$, where $m \ll n$.
> **Given:** $y = AG(x^*) + e \in \mathbb{R}^m$, for some *unknown* latent vector $x^* \in \mathbb{R}^k$, noise vector $e \in \mathbb{R}^m$.
> **Goal:** Recover $x^*$ (or in a different variant of the problem just $G(x^*)$).

It has been shown empirically that this problem (and some variants of it) can be solved efficiently [2]. It has also been shown empirically that the quality of the reconstructed signals in the low number of measurements regime might greatly outperform those reconstructed using a sparsity assumption. It has even been shown that the network $G$ need not be trained on data from the domain of interest $\mathcal{D}$ but that a convolutional neural network $G$ with random weights might suffice to regularize the reconstruction well [18, 19].

Despite the non-convexity of the optimization problem, some theoretical guarantees have also emerged [8, 10], when $G$ is a fully-connected ReLU neural network of the following form (where $d$ is the depth):

$$G(x) = \mathsf{ReLU}(W^{(d)}(\ldots \mathsf{ReLU}(W^{(2)}(\mathsf{ReLU}(W^{(1)}x)))\ldots)), \qquad (1)$$

where each $W^{(i)}$ is a matrix of dimension $n_i \times n_{i-1}$, such that $n_0 = k$ and $n_d = n$. These theoretical guarantees mirror well-known results in sparsity-based compressed sensing, where efficient recovery is possible if the measurement matrix $A$ satisfies a certain deterministic condition, e.g. the Restricted Isometry Property. But for arbitrary $G$, recovery is in general intractable [13], so some assumption about $G$ must also be made. Specifically, it has been shown in prior work that, if the measurement matrix $A$ satisfies a certain *Range Restricted Isometry Condition (RRIC)* with respect to $G$, and each weight matrix $W^{(i)}$ satisfies a *Weight Distribution Condition (WDC)*, then $x^*$ can be efficiently recovered up to error roughly $||e||$ from $O(k \cdot \log(n) \cdot \mathrm{poly}(d))$ measurements [8, 10]. See Section 3 for a definition of the WDC, and Section B for a definition of the RRIC.

But it's critical to understand when these conditions are satisfied (for example, in the sparsity setting, the Restricted Isometry Property is satisfied by i.i.d. Gaussian matrices when $m \geq k \log n$). Similarly, the RRIC has been shown to hold when $A$ is i.i.d. Gaussian and $m \geq k \cdot \log(n) \cdot \mathrm{poly}(d)$, which is an essentially optimal measurement complexity if $d$ is constant. However, until this work, the WDC has seemed more onerous. Under the assumption that each $W^{(i)}$ has i.i.d. Gaussian entries, the WDC was previously only known to hold when $n_i \geq c n_{i-1} \log n_{i-1}$: i.e. when every layer of the neural network is larger than the previous by a logarithmic factor. This *expansivity* condition is a major limitation of the prior theory, since in practice neural networks do not expand at every layer.

Our work alleviates this limitation, settling a problem left open in [8, 10] and recently also posed in survey [15]. We show that the WDC holds when $n_i \geq c n_{i-1}$. This proves the following result, where our contribution is to replace $n_i \geq n_{i-1} \cdot \log(n_{i-1}) \cdot \mathrm{poly}(d)$ with $n_i \geq n_{i-1} \cdot \mathrm{poly}(d)$.

**Theorem 1.1.** *Suppose that each weight matrix has expansion $n_i \geq n_{i-1} \cdot \mathrm{poly}(d)$, and the number of measurements is $m \geq k \cdot \log(n) \cdot \mathrm{poly}(d)$. Suppose that $A$ has i.i.d. Gaussian entries $N(0, 1/m)$ and each $W^{(i)}$ has i.i.d. Gaussian entries $N(0, 1/n_i)$. Then there is an efficient gradient-descent based algorithm which given $G$, $A$, and $y = A \cdot G(x^*) + e$, outputs, with probability at least $1 - e^{-k/\mathrm{poly}(d)}$, an estimate $\tilde{x} \in \mathbb{R}^k$ satisfying $\|\tilde{x} - x^*\| \leq O(2^d \|e\|)$ when $\|e\|$ is sufficiently small.*

We note that the dependence in $d$ of the expansivity, number of measurements, and error in our theorem is the same as in [10]. Moreover, the techniques developed in this paper yield several generalizations of the above theorem, stated informally below, and discussed further in Section F.

**Theorem 1.2.** *Suppose $G$ is a random neural network with constant expansion, and conditions analogous to those of Theorem 1.1 are satisfied. Then the following results also hold with high probability. There is an efficient algorithm for rate-optimal denoising with generative prior $G$. Phase retrieval and one-bit recovery with generative prior $G$ have no spurious local minima. And compressed sensing with a two-layer deconvolutional prior has no spurious local minima.*

To see why expansivity plays a role in the first place, we provide some context:

**Global landscape analysis.** The theoretical guarantees of [8, 10] fall under an emerging method for analyzing non-convex optimization problems called *global landscape analysis* [17]. Given a non-convex function $f$, the basic goal is to show that $f$ does not have spurious local minima, implying that gradient descent will (eventually) converge to a global minimum. Stronger guarantees may

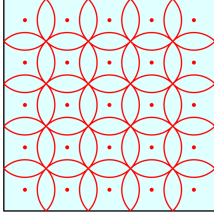 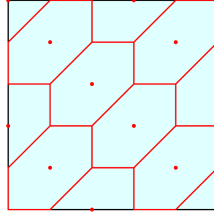 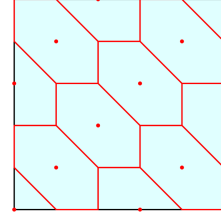

(a) Spherical $\epsilon$-Net: for all $t$, $f_t$ deviates by at most $\epsilon$ within each ball.

(b) Aspherical $\epsilon$-Net: for a *specific* $t$, $f_t$ deviates by at most $\epsilon$ within each weirdly-shaped pseudo-ball.

(c) Aspherical $\epsilon$-Net': for a different $t'$, $f_{t'}$ deviates by at most $\epsilon$ within each weirdly-shaped pseudo-ball.

Figure 1: A parameter-independent spherical $\epsilon$-net, versus a parameter-dependent aspherical $\epsilon$-net for two specific parameters; in particular, the parameter $t$ determines the rotational angle of the weirdly-shaped pseudo-balls within which $f_t$ changes by at most $\epsilon$. The shaded square is $\mathcal{X}$. Each ball in the leftmost panel is the intersection of weirdly-shaped pseudo-balls with the same center, under all possible rotations. As such the radius of this ball is small, so to cover $\mathcal{X}$ with such balls we need a lot of small balls. Instead, the aspherical parameter-dependent $\epsilon$-nets are more efficient.

provide bounds on the convergence rate. In less well-behaved settings, the goal may be to prove guarantees in a restricted region, or show that the local minima inform the global minimum.

In stochastic optimization settings wherein $f$ is a random function, global landscape analysis typically consists of two steps: first, prove that $\mathbb{E}f$ is well-behaved, and second, apply concentration results to prove that, with high probability, $f$ is sufficiently close to $\mathbb{E}f$ that no pathological behavior can arise. The analysis of compressed sensing with generative priors by [8] follows this two-step outline (see Section E for a sketch). The second step requires inducting on the layers of the network. For each layer, it's necessary to prove uniform concentration of a function which takes a weight matrix $W^{(i)}$ as a parameter; this concentration is precisely the content of the WDC. As a general rule, tall random matrices concentrate more strongly, which is why proving the WDC for random matrices requires an expansivity condition (a lower bound on the aspect ratio of each weight matrix).

**Concentration of random functions.** The uniform concentration required for the above analysis can be abstracted as follows. Given a family of functions $\{f_t : \mathcal{X} \to \mathbb{R}\}_{t \in \Theta}$ on some metric space $\mathcal{X}$, and given a distribution $P$ on $\Theta$, we pick a random function $f_t$. We seek to show that with high probability, $f_t$ is uniformly concentrated near $\mathbb{E}f_t$. A generic approach to solve this problem is via Lipschitz bounds. If $f_t$ is sufficiently Lipschitz for all $t \in \Theta$, and $f_t(x)$ is near $\mathbb{E}f_t(x)$ with high probability for any single $x \in \mathcal{X}$, then by union bounding over an $\epsilon$-net, uniform concentration follows.

However, in the global landscape analysis conducted by [8], the functions $f_t$ have poor Lipschitz constants. Pushing through the $\epsilon$-net argument therefore requires very strong pointwise concentration, and hence the expansivity condition is necessitated.

## 1.1 Technical contributions

Concentration of Lipschitz random functions is a widely-used tool in probability theory, which has found many applications in global landscape analysis for the purposes of understanding non-convex optimization, as outlined above. For the functions arising in our analysis, however, Lipschitzness is actually *too strong* a property, and leads to suboptimal results. A main technical contribution of our paper is to define a relaxed notion of *pseudo-Lipschitz* functions and derive a concentration inequality for pseudo-Lipschitz random functions. This serves as a central tool in our analysis, and is a general tool that we envision will find other applications in probability and non-convex optimization.

Informally, a function family $\{f_t : \mathcal{X} \to \mathbb{R}\}_{t \in \Theta}$ is *pseudo-Lipschitz* if for every $t$ there is a *pseudo-ball* such that $f_t$ has small deviations when its argument is varied by a vector within the pseudo-ball. If for every $t$ the pseudo-ball is simply a ball, then the function family is plain Lipschitz. But this definition is more flexible; the pseudo-ball can be any small-diameter, convex body with non-negligible volume and, importantly, every $t$ could have a different pseudo-ball of small deviations. We

show that uniform concentration still holds; here is a simplified (and slightly specialized) statement of our result (presented in full detail in Section 4 along with a formal definition of pseudo-Lipschitzness):

**Theorem 1.3** (Informal-Concentration of pseudo-Lipschitz random functions). *Let $\theta$ be a random variable taking values in $\Theta$. Let $\{f_t : \mathcal{X} \to \mathbb{R}\}_{t \in \Theta}$ be a function family, where $\mathcal{X}$ is a subset of $\mathcal{B}$, the unit-ball in $\mathbb{R}^k$. Suppose that:*

*(1) For any fixed $x$, the random variable $f_\theta(x)$ is well-concentrated around its mean,*

*(2) $\{f_t\}_{t \in \Theta}$ is pseudo-Lipschitz,*

*(3) $\mathbb{E}f_\theta(x)$ is Lipschitz in $x$.*

*Then $f_\theta$ is well-concentrated around its mean, uniformly in $x$. Quantitatively, the strength of the concentration in (1) only needs to be proportional to the inverse volume of the pseudo-balls of small deviation guaranteed by (2) raised to the power $k$, i.e. the number of pseudo-balls needed to cover $\mathcal{B}$.*

This result achieves asymptotically stronger results than are possible through Lipschitz concentration. Where does the gain come from? For each parameter $t$, consider the pseudo-ball of $\epsilon$-deviations of $f_t$, as guaranteed by the pseudo-Lipschitzness. A standard $\epsilon$-net would be covering the metric space by balls (as exemplified in Figure 1a), each of which would have to lie in the intersection of the pseudo-balls of all parameters $t$. If for each parameter the pseudo-ball is "wide" in a different direction (see Figures 1b and 1c for a schematic), then the balls of the standard $\epsilon$-net may be very small compared to any pseudo-ball, and the size of the standard $\epsilon$-net could be very large compared to the size of the $\epsilon$-net obtained for any fixed $t$ using pseudo-balls. Hence, the standard Lipschitzness argument may require much stronger concentration in (1) than our result does, in order to union bound over a potentially much larger $\epsilon$-net.

There is an obvious technical obstacle to our proof: the pseudo-balls depend on the parameter $t$, so an efficient covering of $\mathcal{X}$ by pseudo-balls will necessarily depend on $t$. It's then unclear how to union bound over the centers of the pseudo-balls (as in the standard Lipschitz concentration argument). We resolve the issue with a decoupling argument. Thus, we ultimately show that under mild conditions, the pseudo-Lipschitz random function is asymptotically as well-concentrated as a Lipschitz random function—even though its Lipschitz constant may be much worse.

**Applications.** With our new technique, we are able to show that the WDC holds for $n \times k$ Gaussian matrices whenever $n \geq ck$, for some absolute constant $c$, where previously it was only known to hold if $n \geq ck \log k$. As a consequence, we show Theorem 1.1: that compressed sensing with a random neural network prior does not require the logarithmic expansivity condition.

In addition, there has been follow-up research on variations of the CS-DGP problem described above. The WDC is a critical assumption which enables efficient recovery in the setting of Gaussian noise [9], as well as global landscape analysis in the settings of phaseless measurements [7], one-bit (sign) measurements [16], and two-layer convolutional neural networks [14]. Moreover, there are currently no known theoretical results in this area—compressed sensing with generative priors—that avoid the WDC: hence, until now, logarithmic expansion was necessary to achieve any provable guarantees. Our result extends the prior work on these problems, in a black-box fashion, to random neural networks with constant expansion. We refer to Section F for details about these extensions.

**Lower bound.** As a complementary contribution, we also provide a simple lower bound on the expansion required to recover the latent vector. This lower bound only applies to one-layer neural networks (or, to the input layer of a multi-layer network), but is strong in several other senses: it applies even in the absence of compression and noise, and it is an information-theoretic lower bound. Without compression and noise, the problem is simply inverting a neural network, and it has been shown [13] that inversion is computationally tractable if the network consists of Gaussian matrices with expansion $2 + \epsilon$. In this setting our lower bound is tight: we show that expansion by a factor of 2, at least at the input layer, is in fact necessary for exact recovery. Note, however, that the lower bound applies only to the problem of recovering the latent vector $x^*$, and not to the related problem of recovering the signal $G(x^*)$. Details are deferred to Section C.

## 1.2 Roadmap

In Section 2, we introduce basic notation. In Section 3 we formally introduce the Weight Distribution Condition, and present our main theorem about the Weight Distribution Condition for random matrices. In Section 4 we define pseudo-Lipschitz function families and formalize Theorem 1.3, the main technical result of this paper. In Section 5, we sketch how it implies that Gaussian random matrices with constant expansion satisfy the WDC. Finally, in Section 6 we prove the main technical result. The proof of Theorem 1.1 is deferred to Section B.

Alphabetic sections (A, B, C, etc.) are found in the supplementary material.

## 2 Preliminaries

For any vector $v \in \mathbb{R}^k$, let $\|v\|$ refer to the $\ell_2$ norm of $x$, and for any matrix $A \in \mathbb{R}^{n \times k}$ let $\|A\|$ refer to the operator norm of $A$. If $A$ is symmetric, then $\|A\|$ is also equal to the spectral norm $\lambda_{\max}(A)$. Let $\mathcal{B}$ refer to the unit $\ell_2$ ball in $\mathbb{R}^k$ centered at 0, and let $S^{k-1}$ refer to the corresponding unit sphere. For a set $S \subseteq \mathbb{R}^k$ let $\alpha S = \{\alpha x : x \in S\}$ and let $\beta + S = \{\beta + x : x \in S\}$. For a matrix $W \in \mathbb{R}^{n \times k}$ and a vector $x \in \mathbb{R}^k$, let $W_{+,x}$ be the $n \times k$ matrix $W_{+,x} = \mathrm{diag}(Wx > 0)W$. That is, row $i$ of $W_{+,x}$ is equal to $W_i$ if $W_i x > 0$, and is equal to 0 otherwise.

## 3 Weight Distribution Condition

In the existing literature in compressed sensing, many results for recovery of a sparse signal are based on an assumption on the measurement matrix that is called the Restricted Isometry Property (RIP). Many results then follow the same paradigm: they first prove that sparse recovery is possible under the RIP, and then show that a random matrix drawn from some specific distribution or class of distributions satisfies the RIP. The same paradigm has been followed in the literature of signal recovery under the deep generative prior. In virtually all of these results, the properties that correspond to the RIP property are the combination of the Range Restricted Isometry Condition (RRIC) and the Weight Distribution Condition (WDC). Our main focus in this paper is to improve upon the existing results related to the WDC property. The WDC has the following definition due to [8].

**Definition 3.1.** A matrix $W \in \mathbb{R}^{n \times k}$ is said to satisfy the *(normalized) Weight Distribution Condition (WDC)* with parameter $\epsilon$ if for all nonzero $x, y \in \mathbb{R}^k$ it holds that $\left\| \frac{1}{n} W_{+,x}^T W_{+,y} - Q_{x,y} \right\| \leq \epsilon$, where $Q_{x,y} = \frac{1}{n} \mathbb{E} W_{+,x}^T W_{+,y}$ (with expectation over i.i.d. $N(0,1)$ entries of $W$).

*Remark.* Note that the factor $1/n$ in front of $W_{+,x}^T W_{+,y}$ is not present in the actual condition [8], hence the term "normalized". We scale up $W$ by a factor of $\sqrt{n}$ to simplify later notation.

In Section E, we provide a detailed explanation of why the WDC arises and we also give a sketch of the basic theory of global landscape analysis for compressed sensing with generative priors.

### 3.1 Weight Distribution Condition from constant expansion

To prove Theorem 1.1, our strategy is to prove that the WDC holds for Gaussian random matrices with constant aspect ratio:

**Theorem 3.2.** *There are constants $c, C > 0$ with the following property. Let $\epsilon > 0$ and let $n, k \in \mathbb{N}$. Suppose that $n \geq c \cdot k \cdot \epsilon^{-2} \log(1/\epsilon)$. If $W \in \mathbb{R}^{n \times k}$ is a matrix with independent entries drawn from $N(0,1)$, then $W$ satisfies the normalized WDC with parameter $\epsilon$, with probability at least $1 - e^{-Ck}$.*

Equivalently, if $W$ has entries i.i.d. $N(0, 1/n)$, then it satisfies the unnormalized WDC with high probability. The proof of 3.2 is sketched in Section 5 and provided in full in Section A. It uses concentration of pseudo-Lipschitz functions, which are introduced formally in Section 4. As shown in Section B, Theorem 1.1 then follows from prior work.

## 4 Uniform concentration beyond Lipschitzness

In this section we present our main technical result about uniform concentration bounds. We generalize a folklore result about uniform concentration of Lipschitz functions by generalizing

Lipschitzness to a weaker condition which we call *pseudo-Lipschitzness*. This concentration result can be used to prove Theorem 3.2. Moreover we believe that it may have broader applications.

Before stating our result, let us first define the notion of pseudo-Lipschitzness of function families and give some comparison with the classical notion of Lipschitzness. Let $\{f_t : (\mathbb{R}^k)^d \to \mathbb{R}\}$ be a family of functions over matrices parametrized by $t \in \Theta$. We have the following definitions:

**Definition 4.1.** A set system $\{B_t \subseteq \mathbb{R}^k : t \in \Theta\}$ is $(\delta, \gamma)$-wide if $B_t = -B_t$, $B_t$ is convex, and $\mathrm{Vol}(B_t \cap \delta\mathcal{B}) \geq \gamma \, \mathrm{Vol}(\delta\mathcal{B})$ for all $t \in \Theta$.

**Definition 4.2** (Pseudo-Lipschitzness)**.** Suppose that there exists a $(\delta, \gamma)$-wide set system $\{B_t \subseteq \mathbb{R}^k : t \in \Theta\}$, such that $|f_t(x) - f_t(y)| \leq \epsilon$ for any $t \in \Theta$ and $x, y \in (\mathbb{R}^k)^d$ with $y_i - x_i \in B_t$ for all $i \in [d]$. Then we say that $\{f_t\}_{t \in \Theta}$ is $(\epsilon, \delta, \gamma)$-pseudo-Lipschitz.

**Example 4.3.** *Let $\Theta = \{w \in \mathbb{R}^k : \|w\| \leq 2\sqrt{k}\}$ and let $\epsilon > 0$. Then the family of functions $\{f_w(x) = w \cdot x\}_{w \in \Theta}$ is only $2\sqrt{k}$-Lipschitz. So to have $|f_w(x) - f_w(y)| \leq \epsilon$ we need $\|x - y\| \leq \epsilon/(2\sqrt{k})$. On the other hand, it can be seen that the set system $\{B_w \subseteq \mathbb{R}^k : w \in \Theta\}$ defined by $B_w = \{x \in \mathbb{R}^k : |w \cdot x| \leq \epsilon\}$ is $(c\epsilon, 1/2)$-wide for a constant $c > 0$ (by standard arguments about spherical caps). Therefore the family of functions $f_w(x) = w \cdot x$ is $(\epsilon, c\epsilon, 1/2)$-pseudo-Lipschitz.*

Our main technical result is that the above relaxation of Lipschitzness suffices to obtain strong uniform concentration of measure results; see Section 6 for the proof.

**Theorem 4.4.** *Let $\theta$ be a random variable taking values in $\Theta$. Let $\{f_t : (\mathbb{R}^k)^d \to \mathbb{R} : t \in \Theta\}$ be a function family, and let $g : (\mathbb{R}^k)^d \to \mathbb{R}$ be a function. Let $\epsilon, \gamma, D > 0$ and $\delta \in (0, 1)$. Define the spherical shell $\mathcal{H} = (1 + \delta/2)\mathcal{B} \setminus (1 - \delta/2)\mathcal{B}$ in $\mathbb{R}^k$. Suppose that:*

1. *For any fixed $x \in \mathcal{H}^d$: $\Pr_\theta[f_\theta(x) \leq g(x) + \epsilon] \geq 1 - p$,*

2. *$\{f_t\}_{t \in \Theta}$ is $(\epsilon, \delta, \gamma)$-pseudo-Lipschitz,*

3. *$|g(x) - g(y)| \leq D$ whenever $x \in (S^{k-1})^d$, $y \in (\mathbb{R}^k)^d$, and $\|y_i - x_i\|_2 \leq \delta$ for all $i \in [d]$.*

*Then:* $\Pr_\theta \left[ f_\theta(x) \leq g(x) + 2\epsilon + D, \ \forall x \in (S^{k-1})^d \right] \geq 1 - \gamma^{-2d}(4/\delta)^{2kd}p.$

As a comparison, if the family $\{f_t\}_t$ were simply $L$-Lipschitz, then uniform concentration would hold with probability $1 - (3L/\epsilon)^{kd}p$ by standard arguments. So the "effective Lipschitz constant" of an $(\epsilon, \delta, \gamma)$-pseudo-Lipschitz family is $O(\epsilon/\delta)$ when $\gamma = \Omega(1)$.

# 5 Proof sketch of Theorem 3.2

In [8], a weaker version of Theorem 3.2 was proven—it required a logarithmic aspect ratio (i.e. $n \geq \Omega(k \log k)$). The proof was by a standard $\epsilon$-net argument. In this section we discuss why Theorem 3.2 cannot be proven by standard arguments, and sketch how Theorem 4.4 yields a proof.

Throughout, we let $W$ be an $n \times k$ random matrix with rows $w_1, \ldots, w_n \sim N(0, 1)^k$.

At a high level, the proof of Theorem 3.2 uses an $\epsilon$-net argument, with several crucial twists. The first twist is borrowed from the prior work [8]: we wish to prove concentration for the random function

$$W_{+,x}^T W_{+,y} = \sum_{i=1}^n \mathbb{1}_{w_i^T x > 0} \mathbb{1}_{w_i^T y > 0} w_i w_i^T,$$

but it is not continuous in $x$ and $y$. So for $\epsilon > 0$, define the continuous functions

$$h_{-\epsilon}(z) = \begin{cases} 0 & z \leq -\epsilon, \\ 1 + z/\epsilon & -\epsilon \leq z \leq 0, \\ 1 & z \geq 0. \end{cases} \quad \text{and} \quad h_\epsilon(z) = \begin{cases} 0 & z \leq 0, \\ z/\epsilon & 0 \leq z \leq \epsilon, \\ 1 & z \geq \epsilon. \end{cases}$$

Following [8], we can now define continuous approximations of $W_{+,x}^T W_{+,y}$:

$$G_{W,-\epsilon}(x, y) = \sum_{i=1}^n h_{-\epsilon}(w_i \cdot x) h_{-\epsilon}(w_i \cdot y) w_i w_i^T \quad \text{and} \quad G_{W,\epsilon}(x, y) = \sum_{i=1}^n h_\epsilon(w_i \cdot x) h_\epsilon(w_i \cdot y) w_i w_i^T.$$

Observe that $h_{-\epsilon}$ is an upper approximation to $\mathbb{1}_{z\geq 0}$, and $h_\epsilon$ is a lower approximation. Thus, it's clear that for all $W \in \mathbb{R}^{n\times k}$ and all nonzero $x, y \in \mathbb{R}^k$ we have the matrix inequality

$$G_{W,\epsilon}(x,y) \preceq W_{+,x}^T W_{+,y} \preceq G_{W,-\epsilon}(x,y). \tag{2}$$

So it suffices to upper bound $G_{W,-\epsilon}(x,y)$ and lower bound $G_{W,\epsilon}(x,y)$. The two arguments are essentially identical, and we will focus on the former. We seek to prove that with high probability over $W$, for all $x, y \in S^{k-1}$ simultaneously, $G_{W,-\epsilon}(x,y) \preceq nQ_{x,y} + \epsilon n I_k$. At this point, [8] employs a standard $\epsilon$-net argument. This does not suffice for our purposes, because it uses the following bounds:

(a) For fixed $x, y, u \in S^{k-1}$, the inequality $\frac{1}{n}u^T G_{W,-\epsilon}(x,y)u \leq u^T Q_{x,y}u + \epsilon$ holds with probability $1 - e^{-\Theta(n)}$

(b) $\frac{1}{n}u^T G_{W,-\epsilon}(x,y)u$ is $\Theta(\sqrt{k})$-Lipschitz.

The second bound means that the $\epsilon$-net needs to have granularity $O(1/\sqrt{k})$, so we must union bound over $\sqrt{k}^{O(k)} = e^{O(k\log k)}$ triples $x, y, u \in S^{k-1}$. Thus, a high probability bound requires $n \geq \Omega(k\log k)$. Moreover, both bounds (a) and (b) are asymptotically optimal. So a different approach is needed.

This is where Theorem 4.4 comes in. Let $f_W(x,y,u) = u^T G_{W,-\epsilon}(x,y)u$. If we center a ball of small constant radius at some point $(x, y, u)$, then for any $W$ there is some point in the ball where $f_W$ differs by $\Theta(\sqrt{k})$. But for each $W$, at most points $f_W$ only differs by $\Theta(1)$. More formally, it can be shown that $f_W$ is $(\epsilon, c\epsilon^2, 1/2)$-pseudo-Lipschitz (so its "effective Lipschitz parameter" has no dependence on $k$). The desired concentration is then a corollary of Theorem 4.4.

## 6 Proof of Theorem 4.4

Let $\{B_t\}_{t\in\Theta}$ be a family of sets $B_t \subseteq \mathbb{R}^k$ witnessing that $\{f_t\}_{t\in\Theta}$ is $(\epsilon, \delta, \gamma)$-pseudo-Lipschitz—i.e. $\{B_t\}_{t\in\Theta}$ is $(\delta, \gamma)$-wide, and $|f_t(x) - f_t(y)| \leq \epsilon$ whenever $t \in \Theta$ and $x, y \in (\mathbb{R}^k)^d$ with $y_i - x_i \in B_t$ for all $i \in [d]$. The standard proof technique for uniform concentration is to fix an $\epsilon$-net, and show that with high probability over the randomness $\theta$, every point in the net satisfies the bound. Here instead, we use the pseudo-balls $\{B_t\}_{t\in\Theta}$ to construct a random, aspherical net $\mathcal{N} \subset \mathbb{R}^k$ that depends on $\theta$ and additional randomness. We'll show that with high probability over all the randomness, every point in the net satisfies the bound. In particular we use the following process to construct $\mathcal{N}$:

Let $x_0 = (1, 0, \ldots, 0) \in S^{k-1}$. We define $x_1, x_2, \ldots$ iteratively. For $j \geq 0$, define $C_j \subseteq \mathbb{R}^k$ by

$$C_j = \bigcup_{0\leq i\leq j}\left(x_i + \frac{1}{2}(B_\theta \cap \delta\mathcal{B})\right).$$

For each $j \geq 0$, if $S^{k-1} \subseteq C_j$ then terminate the process. Otherwise, by some deterministic process, pick

$$x_{j+1} \in S^{k-1} \setminus C_j.$$

Let's say that the process terminates at step $j_{\text{last}}$, producing a sequence of random variables $\{x_0, x_1, \ldots, x_{j_{\text{last}}}\}$ (with randomness introduced by $\theta$). Note that $j_{\text{last}}$ is also a random variable.

For each $0 \leq j \leq j_{\text{last}}$, let $y_j$ be the random variable

$$y_j \sim \text{Unif}\left(x_j + \frac{1}{2}(B_\theta \cap \delta\mathcal{B})\right).$$

That is, $y_j$ is a perturbation of $x_j$ by a uniform sample from the aspherical ball. Let $\mathcal{N}$ be the set $\{y_0, y_1, \ldots, y_{j_{\text{last}}}\}$. Observe that $\mathcal{N} \subset \mathcal{H}$, and $S^{k-1}$ is covered by the pseudo-balls $\{y_j + B_\theta\}_j$.

By a volume argument, we can upper bound $|\mathcal{N}| = j_{\text{last}} + 1$.

**Lemma 6.1.** *The size of $\mathcal{N}$ is at most $\gamma^{-1}(5/\delta)^k$.*

*Proof.* For each $0 \leq j \leq j_{\text{last}}$ define an auxiliary set $C_j' \subseteq \mathbb{R}^k$ by $C_j' = x_i + \frac{1}{4}(B_t \cap \delta\mathcal{B})$. We claim that the sets $C_0', \ldots, C_{j_{\text{last}}}'$ are disjoint. Suppose not; then there are some indices $0 \leq j_1 < j_2 \leq j_{\text{last}}$

and some point $z \in \mathbb{R}^k$ such that $z - x_{j_1} \in \frac{1}{4}(B_\theta \cap \delta\mathcal{B})$ and also $z - x_{j_2} \in \frac{1}{4}(B_\theta \cap \delta\mathcal{B})$. It follows from convexity and symmetry of $B_t \cap \delta\mathcal{B}$ that $x_{j_1} - x_{j_2} \in \frac{1}{2}(B_\theta \cap \delta\mathcal{B})$. So $x_{j_2} \in C_{j_1} \subseteq C_{j_2-1}$, contradicting the definition of $x_{j_2}$. So the sets $C'_0, \dots, C'_{j_{\text{last}}}$ are indeed disjoint.

But each $C'_j$ is a subset of $(1 + \delta/4)\mathcal{B}$. By the volume lower bound on $B_t \cap \delta\mathcal{B}$, we have

$$\frac{\text{Vol}(C'_j)}{\text{Vol}((1+\delta/4)\mathcal{B})} \geq \frac{\gamma 4^{-k} \text{Vol}(\delta\mathcal{B})}{\text{Vol}((1+\delta/4)\mathcal{B})} = \gamma(1+4/\delta)^{-k}.$$

The lemma follows. □

We now show that with high probability the inequality $f_\theta(a) \leq g(a) + \epsilon$ holds for all $a \in \mathcal{N}^d$ simultaneously. The main idea is that the random perturbations partially decoupled the net $\mathcal{N}$ from $\theta$. Since each point of the net is distributed uniformly over a set of non-negligible volume, the probability that any fixed $a \in \mathcal{N}^d$ fails the concentration inequality can be bounded against the probability that a uniformly random point from the shell $\mathcal{H}^d$ fails.

**Lemma 6.2.** *We have*

$$\Pr[\exists a \in \mathcal{N}^d : f_\theta(a) > g(a) + \epsilon] \leq \gamma^{-2d}(4/\delta)^{2kd}p.$$

*Proof.* For any $a \in \mathcal{H}^d$ let $E_{\text{bad}}(\theta, a)$ be the event that $f_\theta(a) > g(a) + \epsilon$. Fix $j \in \mathbb{N}^d$. Let $A_j$ be the event that $j_1, \dots, j_d \leq j_{\text{last}}$. (Recall that $j_{\text{last}}$ is a deterministic function of $\theta$ which is random.) Let $X_1, \dots, X_d \sim \text{Unif}(\mathcal{H})$ be independent uniform random variables over the shell $\mathcal{H}$. Let $E_{\text{cont}}$ be the event that $X_i \in x_{j_i} + \frac{1}{2}(B_\theta \cap \delta\mathcal{B})$ for all $i \in [d]$, where for convenience we define $x_j = (1, 0, \dots, 0) \in S^{k-1}$ for $j > j_{\text{last}}$. For any $t \in \Theta$, consider conditioning on $(\theta = t)$. Then $j_{\text{last}}$ and $x_0, \dots, x_{j_{\text{last}}}$ are deterministic; assume that $A_j$ occurs. The conditional random vector $(y_{j_1}, \dots, y_{j_d})|(\theta = t)$ has the uniform product distribution

$$\text{Unif}\left(\prod_{i=1}^d \left(x_{j_1} + \frac{1}{2}(B_t + \delta\mathcal{B})\right)\right).$$

This is precisely the distribution of $(X_1, \dots, X_d)|(E_{\text{cont}}, \theta = t)$. Thus,

$$\Pr[E_{\text{bad}}(\theta, (y_{j_1}, \dots, y_{j_d}))|\theta = t] = \Pr[E_{\text{bad}}(\theta, (X_1, \dots, X_d))|E_{\text{cont}}, \theta = t]$$
$$\leq \frac{\Pr[E_{\text{bad}}(\theta, (X_1, \dots, X_d))|\theta = t]}{\Pr[E_{\text{cont}}|\theta = t]}. \qquad (3)$$

Since $(X_1, \dots, X_d)$ are independent and uniformly distributed over $\mathcal{H}$,

$$\Pr[E_{\text{cont}}|\theta = t] = \left(\frac{\text{Vol}((B_t \cap \delta\mathcal{B})/2)}{\text{Vol}(\mathcal{H})}\right)^d \geq \left(\frac{2^{-k}\gamma \text{Vol}(\delta\mathcal{B})}{\text{Vol}((1+\delta/2)\mathcal{B})}\right)^d \geq \gamma^d(\delta/3)^{kd}.$$

Substituting into Equation (3) and integrating over all $\theta \in A_j$,

$$\Pr[E_{\text{bad}}(\theta, (y_{j_1}, \dots, y_{j_d})) \wedge A_j] \leq \gamma^{-d}(3/\delta)^{kd} \Pr[E_{\text{bad}}(\theta, (X_1, \dots, X_d)) \wedge A_j].$$

If $(X_1, \dots, X_d)$ were deterministic then we would have $f_\theta(X_1, \dots, X_d) \leq g(X_1, \dots, X_d)]$ with probability at least $1 - p$. They are not deterministic, but they are independent of $\theta$, which suffices to imply the above inequality. So

$$\Pr[E_{\text{bad}}(\theta, (y_{j_1}, \dots, y_{j_d})) \wedge A_j] \leq \gamma^{-d}(3/\delta)^{kd}p.$$

Finally we take a union bound over $j$. By Lemma 6.1 we have $j_{\text{last}} < \gamma^{-1}(5/\delta)^k$. So

$$\Pr[\exists a \in \mathcal{N}^d : E_{\text{bad}}(\theta, a)] \leq (\gamma^{-1}(5/\delta)^k)^d \gamma^{-d}(3/\delta)^{kd}p \leq \gamma^{-2d}(4/\delta)^{2kd}p.$$

as claimed. □

We conclude the proof of Theorem 4.4. Suppose that the event of Lemma 6.2 fails; that is, for all $a \in \mathcal{N}^d$ we have $f_\theta(a) \le g(a) + \epsilon$. Then let $b \in (S^{k-1})^d$. By construction of $\mathcal{N}$, for every $i \in [d]$ there is some $0 \le j_i \le j_{\text{last}}$ such that $b_i \in x_{j_i} + \frac{1}{2}(B_\theta \cap \delta\mathcal{B})$. But $y_{j_i} \in x_{j_i} + \frac{1}{2}(B_\theta \cap \delta\mathcal{B})$, so by convexity and symmetry of $B_\theta$, it follows that $b_i - y_{j_i} \in B_\theta$. Hence by assumption, we have $|f_\theta(b) - f_\theta(y_{j_1}, \ldots, y_{j_d})| \le \epsilon$. Since $(y_{j_1}, \ldots, y_{j_d}) \in \mathcal{N}^d$, we have

$$f_\theta(y_{j_1}, \ldots, y_{j_d}) \le g(y_{j_1}, \ldots, y_{j_d}) + \epsilon.$$

Thus,

$$f_\theta(b) \le f_\theta(y_{j_1}, \ldots, y_{j_d}) + \epsilon \le g(y_{j_1}, \ldots, y_{j_d}) + 2\epsilon \le g(b) + 2\epsilon + D$$

as desired. By Lemma 6.2, this uniform bound holds with probability at least $1 - \gamma^{-2d}(4/\delta)^{2kd}p$, over the randomness of $\theta$ and the net perturbations. So it also holds with at least this probability over just the randomness of $\theta$.

## Broader Impact

Our main contributions are mathematical in nature. We establish the notion of *pseudo-Lipschitzness*, along with a concentration inequality for random pseudo-Lipschitz functions, and random matrices, and we use our results to further the theoretical understanding of the non-convex optimization landscape arising in compressed sensing with deep generative priors. We foresee applications of our theorems in probability theory, learning theory, as well as inverse optimization problems involving deep neural networks. That said, compressed sensing with deep generative priors is of practical relevance as well. As shown in recent work, in the low number of measurements regime, compressed sensing with a deep generative prior may significantly outperform compressed sensing with a sparsity assumption. We emphasize, however, that users of a deep generative prior in compressed sensing (or other inverse problems) should be cognizant of the risk that the prior may introduce bias in the reconstruction. Indeed, the deep generative model was trained on data which might be biased, and even if it is not biased the training of the deep generative model might have failed for statistical or optimization reasons, resulting in a biased trained model. So the reconstruction will only be as good as the deep generative model is, as the reconstructed signal is in the range of the deep generative model. To conclude, our contributions are methodological but a prime application of our techniques is to improve the understanding of optimization problems arising in inverse problems involving a deep generative model. The users of deep generative priors in practical scenarios must be careful about the potential biases that their priors introduce. Their provenance and quality must be understood.

## Acknowledgments

We thank Paul Hand and Alex Dimakis for useful discussions. This research was supported by NSF Awards IIS-1741137, CCF-1617730 and CCF-1901292, by a Simons Investigator Award, by the DOE PhILMs project (No. DE-AC05-76RL01830), by the DARPA award HR00111990021, by the MIT Undergraduate Research Opportunities Program, and by a Google PhD Fellowship.

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
