[Supplementary Material]

# supplementary material for paper: Constant-Expansion Suffices for Compressed Sensing with Generative Priors

**Constantinos Daskalakis**
MIT
costis@mit.edu

**Dhruv Rohatgi**
MIT
drohatgi@mit.edu

**Manolis Zampetakis**
MIT
mzampet@mit.edu

## A  Extending WDC to constant expansion (Proof of Theorem 3.2)

In this section we prove Theorem 3.2. Fix $\epsilon > 0$. Let $W \in \mathbb{R}^{n \times k}$ have rows $w_1, \dots, w_n \sim N(0,1)^k$. Recall from Section 5 that we have the matrix inequality

$$G_{W,\epsilon}(x,y) \preceq W_{+,x}^T W_{+,y} \preceq G_{W,-\epsilon}(x,y). \tag{4}$$

So it suffices to upper bound $G_{W,-\epsilon}(x,y)$ and lower bound $G_{W,\epsilon}(x,y)$. The two arguments are essentially identical, and we will focus on the former. We seek to prove that with high probability over $W$, for all $x, y \in S^{k-1}$ simultaneously,

$$G_{W,-\epsilon}(x,y) \preceq nQ_{x,y} + \epsilon n I_k.$$

Moreover we want this to hold whenever $n \geq Ck$ for some constant $C = C(\epsilon)$. We'll use two standard concentration inequalities:

**Lemma A.1.** *Suppose that $k \leq n$. Then*

1. *$\|W\| \leq 3\sqrt{n}$ with probability at least $1 - e^{-n/2}$.*

2. *$\max_{i \in [n]} \|w_i\|_2 \leq \sqrt{2k}$ with probability at least $1 - ne^{-k/8}$.*

*Proof.* See [20] for a reference on the first bound. The second bound is by concentration of chi-squared with $k$ degrees of freedom. $\square$

Let $\Theta$ be the set of matrices $M \in \mathbb{R}^{n \times k}$ such that $\|M\| \leq 3\sqrt{n}$ and $\max_{i \in [n]} \|M_i\|_2 \leq \sqrt{2k}$. Define the random variable $\theta = W | (W \in \Theta)$.

Fix $u \in S^{k-1}$. For any $M \in \Theta$, define

$$f_M(x,y) = \frac{1}{n} u^T G_{M,-\epsilon}(x,y) u$$

and define $g(x,y) = u^T Q_{x,y} u$. We check that $f$ and $g$ satisfy the three conditions of Theorem 4.4 with appropriate parameters.

**Lemma A.2.** *For any $x, y \in \mathbb{R}^k$ with $\|x\|_2, \|y\|_2 \geq 1/2$,*

$$\Pr[f_\theta(x,y) \leq g(x,y) + 3\epsilon] \geq 1 - 4\exp(-Cn\epsilon^2).$$

*Proof.* We first consider $f_W(x,y)$. It is shown in the proof of Lemma 12 in [8] that

$$\mathbb{E}[G_{W,-\epsilon}(x,y)] \preceq Q_{x,y} + \left( \frac{\epsilon}{2\|x\|_2} + \frac{\epsilon}{2\|y\|_2} \right) I,$$

which under our assumptions implies that $\mathbb{E}[f_W(x,y)] \le g(x,y) + 2\epsilon$. Now expand

$$f_W(x,y) = \frac{1}{n} \sum_{i=1}^{n} h_{-\epsilon}(w_i x) h_{-\epsilon}(w_i y)(w_i u)^2.$$

Let $z_i = h_{-\epsilon}(w_i x) h_{-\epsilon}(w_i y)(w_i u)^2$. Since $h_{-\epsilon}(w_i x) h_{-\epsilon}(w_i y)$ is bounded and $w_i u$ is Gaussian, and $\mathbb{E} z_i$ is bounded by a constant, it follows that $z_i - \mathbb{E} z_i$ is subexponential with some constant variance proxy $\sigma$. Therefore by Bernstein's inequality, for all $t > 0$,

$$\Pr[f_W(x,y) - \mathbb{E} f_W(x,y) > t] \le 2\exp(-Cn\min(t,t^2))$$

for some constant $C > 0$. Taking $t = \epsilon$, we get that

$$\Pr[f_W(x,y) > g(x,y) + 3\epsilon] \le 2\exp(-Cn\epsilon^2).$$

Finally, since $\Pr[W \in \Theta] \ge 1/2$, it follows that conditioning on $\Theta$ at most doubles the failure probability. So

$$\Pr[f_\theta(x,y) > g(x,y) + 3\epsilon] \le 4\exp(-Cn\epsilon^2)$$

as desired. $\qquad\square$

Next, we show that $\{f_M\}_{M\in\Theta}$ is $(\epsilon,\delta,\gamma)$-pseudo-Lipschitz where $\delta = \epsilon^2/116$ and $\gamma = 1/2$.

**Definition A.3.** For any $M \in \mathbb{R}^{n\times k}$, $t \in \mathbb{R}$, and $u \in \mathbb{R}^k$, let $B_{M,t,u} \subseteq \mathbb{R}^k$ be the set of points $v \in \mathbb{R}^k$ such that

$$\sum_{i=1}^{n} |M_i v|(M_i u)^2 \le tn.$$

The pseudo-ball $B_{M,t,u}$ captures the directions in which $f_M$ is Lipschitz more effectively than any spherical ball, as the following lemma shows.

**Lemma A.4.** Let $M \in \mathbb{R}^{n\times k}$. Let $x,y,\tilde{x},\tilde{y} \in \mathbb{R}^k$. If $y - \tilde{y} \in B_{M,\epsilon^2/4,u}$ and $x - \tilde{x} \in \mathcal{B}_{M,\epsilon^2/4,u}$ then

$$|f_M(x,y) - f_M(\tilde{x},\tilde{y})| \le \epsilon.$$

*Proof.* We have

$$|f_M(x,y) - f_M(\tilde{x},\tilde{y})| \le \frac{1}{n}\sum_{i=1}^{n} |h_{-\epsilon}(M_i x)h_{-\epsilon}(M_i y) - h_{-\epsilon}(M_i \tilde{x})h_{-\epsilon}(M_i \tilde{y})|(M_i u)^2$$

$$\le \frac{1}{n}\sum_{i=1}^{n} \Big[ h_{-\epsilon}(M_i x)|h_{-\epsilon}(M_i y) - h_{-\epsilon}(M_i \tilde{y})| +$$

$$h_{-\epsilon}(M_i \tilde{y})|h_{-\epsilon}(M_i x) - h_{-\epsilon}(M_i \tilde{x})| \Big](M_i u)^2$$

$$\le \frac{1}{n}\sum_{i=1}^{n} \big[|h_{-\epsilon}(M_i y) - h_{-\epsilon}(M_i \tilde{y})| + |h_{-\epsilon}(M_i x) - h_{-\epsilon}(M_i \tilde{x})|\big](M_i u)^2$$

$$\le \frac{1}{\epsilon n}\sum_{i=1}^{n} \big[|M_i(y - \tilde{y})| + |M_i(x - \tilde{x})|\big](M_i u)^2$$

$$\le \epsilon$$

where the second-to-last inequality uses that $h_{-\epsilon}$ is $1/\epsilon$-Lipschitz, and the last inequality uses the assumptions on $y - \tilde{y}$ and $x - \tilde{x}$. $\qquad\square$

Next, we need to lower bound the volume of $B_{M,t,u}$.

**Lemma A.5.** Fix $u \in S^{k-1}$ and $\delta, t > 0$. Then $\{B_{M,t,u}\}_{M\in\theta}$ is $(\delta,\gamma)$-wide for $\gamma = \delta^k(1 - 41\delta t^{-1})$. Fix $M \in \Theta$ and $u \in S^{k-1}$. For any $\delta, t > 0$,

$$\mathrm{Vol}(B_{M,t,u} \cap \delta\mathcal{B}) \ge \delta^k(1 - 41\delta t^{-1})\,\mathrm{Vol}(\mathcal{B}).$$

*Proof.* Fix $M \in \Theta$. It's clear from the definition that $B_{M,t,u}$ is symmetric (i.e. $v \in B_{M,t,u}$ implies $-v \in B_{M,t,u}$) and convex (by the triangle inequality). It remains to show that

$$\mathrm{Vol}(B_{M,t,u} \cap \delta \mathcal{B}) \geq \delta^k (1 - 41\delta t^{-1}) \, \mathrm{Vol}(\mathcal{B}).$$

Writing the $\mathrm{Vol}(B_{M,t,u})$ as a probability,

$$\frac{\mathrm{Vol}(B_{M,t,u} \cap \delta \mathcal{B})}{\mathrm{Vol}(\mathcal{B})} = \delta^k \frac{\mathrm{Vol}(B_{M,t,u} \cap \delta \mathcal{B})}{\mathrm{Vol}(\delta \mathcal{B})}$$

$$= \delta^k \Pr_{\eta \in \delta \mathcal{B}} \left[ \sum_{i=1}^n |M_i \eta| (M_i u)^2 \leq tn \right].$$

Since increasing $\|\eta\|$ only increases the sum, the probability over $\delta \mathcal{B}$ is at least the probability over the sphere $\{\eta \in \mathbb{R}^k : \|\eta\|_2 = \delta\}$. Then

$$\Pr_{\eta \in \delta \mathcal{B}} \left[ \sum_{i=1}^n |M_i \eta| (M_i u)^2 \leq tn \right] \geq \Pr_{\|\eta\|_2 = \delta} \left[ \sum_{i=1}^n |M_i \eta| (M_i u)^2 \leq tn \right]$$

$$= \Pr_{\eta \sim N(0,1)^k} \left[ \sum_{i=1}^n |M_i \eta| (M_i u)^2 \leq tn \|\eta\|_2 / \delta \right]$$

by spherical symmetry of the Gaussian. Now we upper bound the probability of the complementary event as

$$\Pr_{\eta \sim N(0,1)^k} \left[ \sum_{i=1}^n |M_i \eta| (M_i u)^2 > tn \|\eta\|_2 / \delta \right]$$

$$= \Pr_{\eta \sim N(0,1)^k} \left[ \sum_{i=1}^n |M_i \eta| (M_i u)^2 > tn \|\eta\|_2 / \delta \,\middle|\, \|\eta\|_2 \geq \sqrt{k}/2 \right]$$

$$= \frac{\Pr_{\eta \sim N(0,1)^k} \left[ \sum_{i=1}^n |M_i \eta| (M_i u)^2 > tn \|\eta\|_2 / \delta \wedge \|\eta\|_2 \geq \sqrt{k}/2 \right]}{\Pr[\|\eta\|_2 \geq \sqrt{k}/2]}$$

$$\leq \frac{\Pr_{\eta \sim N(0,1)^k} \left[ \sum_{i=1}^n |M_i \eta| (M_i u)^2 > tn\sqrt{k}/(2\delta) \right]}{1 - e^{-9k/128}}$$

$$\leq 2 \Pr_{\eta \sim N(0,1)^k} \left[ \sum_{i=1}^n |M_i \eta| (M_i u)^2 > tn\sqrt{k}/(2\delta) \right]$$

For each $i \in [n]$, the random variable $M_i \eta$ is distributed as $N(0, \|M_i\|^2)$. Since $M \in \Theta$, we have $\|M_i\| \leq \sqrt{2k}$. Thus, $|M_i \eta|$ has mean at most $\|M_i\| \sqrt{2/\pi} \leq 2\sqrt{k/\pi}$. So by Markov's inequality,

$$\Pr_{\eta \sim N(0,1)^k} \left[ \sum_{i=1}^n |M_i \eta| (M_i u)^2 > tn\sqrt{k}/(2\delta) \right] \leq \frac{\mathbb{E}_{\eta \sim N(0,1)^k} \left[ \sum_{i=1}^n |M_i \eta| (M_i u)^2 \right]}{tn\sqrt{k}/(2\delta)}$$

$$\leq \frac{\sum_{i=1}^n (M_i u)^2 2\sqrt{k/\pi}}{tn\sqrt{k}/(2\delta)}$$

$$\leq 36\delta/(t\sqrt{\pi})$$

where the last inequality uses the assumption $M \in \Theta$ to get $\|Mu\|^2 \leq 9n \|u\|^2 = 9n$. We conclude that

$$\frac{\mathrm{Vol}(B_{M,t,u} \cap \delta \mathcal{B})}{\mathrm{Vol}(\mathcal{B})} \geq \delta^k (1 - 72\delta/(t\sqrt{\pi}))$$

as desired. □

Taking $t = \epsilon^2$ and $\delta = \epsilon^2/82$ yields

$$\mathrm{Vol}(B_{M,t,u} \cap \delta \mathcal{B}) \geq \frac{1}{2} \mathrm{Vol}(\delta \mathcal{B}).$$

So $\{f_M\}_{M\in\Theta}$ is $(2\epsilon, \epsilon^2/82, 1/2)$-pseudo-Lipschitz. Finally, we have a lemma about the smoothness of $g$; see Appendix D for the proof.

**Lemma A.6.** *Let $x, y, \tilde{x}, \tilde{y} \in (3/2)\mathcal{B} \setminus (1/2)\mathcal{B}$. Let $d = \max(\|x - y\|, \|\tilde{x} - \tilde{y}\|)$. Then*

$$\|Q_{x,y} - Q_{\tilde{x},\tilde{y}}\| \le O(\sqrt{\|x - \tilde{x}\|_2 + \|y - \tilde{y}\|_2}).$$

Thus, we have $|g(x, y) - g(\tilde{x}, \tilde{y})| \le C\epsilon$ whenever $x, y \in S^{k-1}$ and $\tilde{x} - x, \tilde{y} - y \in \delta\mathcal{B}$. Having shown that the three conditions of Theorem 4.4 are satisfied, we can prove uniform concentration over all $x, y \in S^{k-1}$.

**Lemma A.7.** *Let $\epsilon > 0$. Fix $u \in S^{k-1}$. Then with probability at least $1 - (C/\epsilon)^{8k}e^{-cn\epsilon^2}$ (over $\theta$), it holds that for all $x, y \in S^{k-1}$,*

$$f_\theta(x, y) \le g(x, y) + C\epsilon n.$$

*Proof.* Apply Theorem 4.4 to the family $\{f_M\}_{M\in\Theta}$ and random variable $\theta$. By Lemma A.2, we can take $p = \exp(-cn\epsilon^2)$ for a constant $c > 0$. We know that $\{f_{x,y}\}$ is $(2\epsilon, \epsilon^2/82, 1/2)$-pseudo-Lipschitz. By Lemma D.4, we can take $D = C\epsilon$. The claim follows. $\qquad\square$

Now we get a uniform bound over all $u \in S^{k-1}$, via a standard $\epsilon$-net an Lipschitz bound. Adding notation for clarity, define

$$f_M(x, y, u) = \frac{1}{n}u^T G_{M,-\epsilon}(x, y)u.$$

The following lemma shows that $f$ is Lipschitz in $u$.

**Lemma A.8.** *Let $M \in \Theta$. Let $x, y \in \mathbb{R}^k$ and $u, v \in S^{k-1}$. Then*

$$|f_M(x, y, u) - f_M(x, y, v)| \le 18n\|u - v\|_2$$

*Proof.* We have

$$
\begin{aligned}
|f_M(x, y, u) - f_M(x, y, v)| &\le \sum_{i=1}^{n} h_{-\epsilon}(M_i x)h_{-\epsilon}(M_i y)\left|(M_i u)^2 - (M_i v)^2\right| \\
&\le \sum_{i=1}^{n} |M_i(u - v)| \cdot |M_i(u + v)| \\
&\le \|M(u - v)\|_2 \|M(u + v)\|_2 \\
&\le \|M\|^2 \|u - v\|_2 \|u + v\|_2 \\
&\le 18n\|u - v\|_2
\end{aligned}
$$

by an application of Cauchy-Schwarz, and the bound $\|M\| \le 3\sqrt{n}$. $\qquad\square$

The matrix bound on $G_{W,-\epsilon}(x, y)$ follows from applying an $\epsilon$-net together with Lemma A.8, and finally removing the conditioning on $\Theta$.

**Theorem A.9.** *Let $\epsilon > 0$. Then with probability at least $1 - (C/\epsilon)^{8k}e^{-cn\epsilon^2} - ne^{-k/8}$ over $W$, it holds that for all $x, y \in S^{k-1}$,*

$$\frac{1}{n}G_{W,-\epsilon}(x, y) \preceq Q_{x,y} + \epsilon.$$

*Proof.* Let $\mathcal{E} \subset S^{k-1}$ be an $\epsilon$-net of size at most $(3/\epsilon)^k$. By a union bound, it holds with probability at least $1 - (3/\epsilon)^k(C/\epsilon)^{8k}e^{-cn\epsilon^2}$ over $\theta$ that for all $x, y \in S^{k-1}$ and all $u \in \mathcal{E}$,

$$f_\theta(x, y, u) \le g(x, y, u) + 2\epsilon.$$

For any $x, y, u \in S^{k-1}$ there is some $v \in \mathcal{E}$ with $\|u - v\|_2 \le \epsilon$, so by Lemma A.8,

$$f_\theta(x, y, u) \le f_\theta(x, y, v) + O(\epsilon) \le g(x, y, v) + O(\epsilon).$$

But $\|Q_{x,y}\| \leq 1$, so

$$|g(x, y, u) - g(x, y, v)| = |u^T Q_{x,y} u - v^T Q_{x,y} v| \leq 2 \|u - v\|_2.$$

Thus, $f_\theta(x, y, u) \leq g(x, y, u) + O(\epsilon)$. We conclude that with probability at least $1 - (3/\epsilon)^k (C/\epsilon)^{8k} e^{-cn\epsilon^2}$ over $\theta$ we have the desired inequality. But $\Pr[W \in \Theta] \geq 1 - ne^{-k/8} - e^{-n/2}$. So the desired inequality holds with probability at least $1 - (3/\epsilon)^k (C/\epsilon)^{8k} e^{-cn\epsilon^2} - ne^{-k/8} - e^{-n/2}$ over $W$. $\qquad\square$

We turn to the lower bound on $G_{W,\epsilon}(x, y)$. The proof is essentially identical. For fixed $x, y$ there is an analogous bound to Lemma A.2:

**Lemma A.10.** *[8] There are constants $c_K, \gamma_K$ such that if $n > c_K \epsilon^{-2} k$ then for any fixed $x, y \in S^{k-1}$ we have that*

$$G_\epsilon(x, y) \succeq nQ_{x,y} - 2\epsilon n I_k$$

*with probability at least $1 - 2e^{-\gamma_K \epsilon^2 n}$.*

The same pseudo-Lipschitz bounds hold for $u^T G_{M,\epsilon}(x, y)u$ as we proved for $u^T G_{M,-\epsilon} u$, and the remaining argument is unchanged. Thus, we have the following theorem.

**Theorem A.11.** *There is a constant $c$ such that if $n > ck\epsilon^{-2} \log(\epsilon^{-2})$ then with high probability over $W$, for all $x, y \in S^{k-1}$,*

$$G_{W,\epsilon}(x, y) \succeq nQ_{x,y} - \epsilon n.$$

Our main result immediately follows.

**Proof of Theorem 3.2.** Recall that for all $x, y \in S^{k-1}$,

$$G_{W,\epsilon}(x, y) \preceq W_{+,x}^T W_{+,y} \preceq G_{W,-\epsilon}(x, y)$$

It follows from Theorems A.9 and A.11 that with high probability over $W$, for all $x, y \in S^{k-1}$,

$$nQ_{x,y} - \epsilon n \preceq W_{+,x}^T W_{+,y} \preceq nQ_{x,y} + \epsilon n.$$

Since $Q_{x,y}$ and $W_{+,x}^T W_{+,y}$ are both invariant under scaling by a positive constant, this inequality then holds for all nonzero $x, y \in \mathbb{R}^k$. So $W$ satisfies the normalized WDC with parameter $\epsilon$. $\qquad\square$

# B   Proof of Main Result: Theorem 1.1

In this section, we prove Theorem 1.1. We apply Theorem 3.2 together with a theorem from [10]. To provide the precise statement we need to introduce several more definitions. First, we introduce the *unnormalized* WDC (as defined in [8]):

**Definition B.1.** A matrix $W \in \mathbb{R}^{n \times k}$ is said to satisfy the *unnormalized Weight Distribution Condition (WDC)* with parameter $\epsilon$ if for all nonzero $x, y \in \mathbb{R}^k$ it holds that

$$\left\| W_{+,x}^T W_{+,y} - Q_{x,y} \right\| \leq \epsilon$$

where $Q_{x,y} = \mathbb{E} W_{+,x}^T W_{+,y}$ (with expectation over i.i.d. $N(0, 1)$ entries of $W$).

Comparing with Definition 3.1, it's clear that $W$ satisfies the unnormalized WDC if and only if $\sqrt{n} W$ satisfies the normalized WDC. In this paper we proved results about the normalized WDC for ease of notation, but for compressed sensing we are concerned that our weight matrices satisfy the unnormalized WDC.

**Definition B.2** ([8])**.** A matrix $A \in \mathbb{R}^{m \times n}$ satisfies the *Range Restricted Isometry Condition* (RRIC) with respect to $G$ and with parameter $\epsilon$ if for all $x, y, z, w \in \mathbb{R}^k$, it holds that

$$|(G(x) - G(y))^T A^T A (G(z) - G(w)) - (G(x) - G(y))^T (G(z) - G(w))|$$
$$\leq \epsilon \|G(x) - G(y)\|_2 \|G(z) - G(w)\|_2.$$

**Definition B.3** ([10])**.** The *empirical risk function* is $f : \mathbb{R}^k \to \mathbb{R}$ defined by

$$f(x) = \frac{1}{2} \|AG(x) - y\|_2^2.$$

Let ALGO refer to Algorithm 1 in [10]. This algorithm is a modification of gradient descent on the empirical risk, which given an initial point $x_0$ computes iterates $x_1, x_2, \ldots$. The result of [10] is a bound on the convergence rate of ALGO, assuming that the (unnormalized) WDC and RRIC are satisfied. We restate it below. We treat network depth $d$ as a constant for simplicity of notation; we denote all upper-bound constants by $C$.

**Theorem B.4** ([10]). *Suppose that $W^{(1)}, \ldots, W^{(d)}$ satisfy the (unnormalized) WDC, and $A$ satisfies the RRIC with respect to $G$, each with parameter $\epsilon \leq C$. Suppose that the noise $e$ satisfies $\|e\| \leq C \|x^*\|$. Let $x_0$ be the initial point of ALGO. Then after $N \leq C f(x_0)/\|x^*\| +$ iterations, the iterate $x_N$ satisfies*

$$\|x_N - x^*\|_2 \leq C(\|x^*\| \sqrt{\epsilon} + \|e\|)$$

*Additionally, for any $i \geq N$,*

$$\|x_{i+1} - x^*\| \leq \gamma^{i+1-N} \|x_N - x^*\| + C \|e\|$$

*where $\gamma \in (0, 1)$ is a constant.*

So let $\epsilon \in (0, C)$. We only need to show that the (unnormalized) WDC and RRIC are satisfied under the conditions of Theorem 1.1. That is, we have the following assumptions (recall that $k = n_0$ and $n = n_d$) for constants $K_2, K_3$:

(a) For $i \in [d]$, the weight matrix $W^{(i)}$ has dimension $n_i \times n_{i-1}$ with

$$n_i \geq K_2 n_{i-1} \epsilon^{-2} \log(1/\epsilon^2).$$

Additionally, the entries are drawn i.i.d. from $N(0, 1/n_i)$.

(b) The measurement matrix $A$ has dimension $m \times n$ with $m \geq K_3 \epsilon^{-1} \log(1/\epsilon) dk \log \prod_{i=1}^{d} n_i$. Additionally, the entries are drawn i.i.d. from $N(0, 1/m)$.

By Theorem 3.2, with high probability, the weight matrices $\sqrt{n_1} W^{(1)}, \ldots, \sqrt{n_d} W^{(d)}$ satisfy the normalized WDC, so $W^{(1)}, \ldots, W^{(d)}$ satisfy the unnormalized WDC. To show that $A$ satisfies the RRIC with respect to $G$, we appeal to Lemma 17 in [8], which proves this exact fact (without any assumptions about the expansion of $G$).

Thus, Theorem B.4 is applicable. This completes the proof of Theorem 1.1.

## C   Lower bound: necessity of non-trivial expansivity

Our main result implied that constant expansion at every layer is sufficient to efficiently recover a latent parameter, even in the presence of compressive measurements and noise. In this section we prove a lower bound: that a factor-of-2 expansion in the *input* layer of the neural network is necessary to recover the latent parameter, even in the absence of compressive measurements and noise. That is, inverting a neural network where the input layer has expansion of less than 2 is impossible.

Note that this lower bound is against recovering the latent parameter $x^*$ from measurements $y = AG(x^*)$ (when $A$ is the identity matrix, and therefore also when $A$ is compressive). Expansivity may not be necessary to recover the actual signal $G(x^*)$.

To prove this lower bound it suffices to consider a one-layer neural network

$$G(x) = \mathsf{ReLU}(Wx),$$

where $x \in \mathbb{R}^k$ and $W \in \mathbb{R}^{m \times k}$. We show that if $m \leq 2k - 1$ then for any weight matrix $W$, there is some signal which corresponds to multiple latent vectors. Moreover if the rows of $W$ have bounded $\ell_2$ norm, then even approximate recovery is impossible.

Note that this bound is tight, since if the rows of $W$ consist of the $2k$ signed basis vectors $\{\pm e_1, \ldots, \pm e_k\}$ then $G$ is injective.

**Proposition C.1.** *Suppose that $m \leq 2k - 1$ and $\max_{i \in [m]} \|W_i\|_2 \leq B$. Then there are $x, y \in \mathbb{R}^k$ with $G(x) = G(y)$ but $\|x - y\|_2 \geq 1/B$.*

*Proof.* If $\text{rank}(W) < k$, then there are distinct $x, y \in \mathbb{R}^k$ with $Wx = Wy$, so certainly $G(x) = G(y)$. Otherwise, the rows of $W$ contain a basis for $\mathbb{R}^k$. Without loss of generality, assume that $W_1, \ldots, W_k$ span $\mathbb{R}^k$. Define $S = \{1, \ldots, k\}$ and $T = \{k+1, \ldots, m\}$. Then the square submatrix $W_S$ is invertible. Define

$$x = (W_S)^{-1} \begin{bmatrix} -1 \\ -1 \\ \vdots \\ -1 \end{bmatrix}.$$

Since $W_T$ has at most $k-1$ rows, it has non-trivial null space. Let $v \in \mathbb{R}^k$ be a unit vector with $W_T v = 0$. Let $\lambda = \|W_S v\|_\infty$. Since $W_S$ is invertible it's clear that $\lambda > 0$. On the other hand $\lambda \le \sup_{i \in [m]} |W_i v| \le B$. Define

$$y = x + v/\lambda.$$

Certainly $\|x - y\|_2 = \lambda^{-1} \ge 1/B$. For any $i \in T$, it's clear that $W_i x = W_i y$. Moreover, fix any $i \in S$. Then $W_i x = -1$, and

$$W_i y = W_i x + W_i v/\lambda \le -1 + \|W_S v\|_\infty / \lambda \le 0.$$

Thus, $\mathsf{ReLU}(W_i x) = \mathsf{ReLU}(W_i y)$ for all $i \in [m]$. So $G(x) = G(y)$. $\qquad\square$

# D  Lipschitz bound for $Q_{x,y}$

In this appendix we show that $Q_{x,y} = \frac{1}{n}\mathbb{E}W_{+,x}^T W_{+,y}$ is Lipschitz (under the operator norm) as a function of $x$ and $y$.

For $x \in \mathbb{R}^k$ nonzero, let $\hat{x}$ refer to the unit vector $x/\|x\|_2$. For $x, y \in \mathbb{R}^k$ nonzero, define $M_{x,y} \in \mathbb{R}^{k \times k}$ to be the matrix such that

$$M_{x,y}(a\hat{x} + b\hat{y} + z) = a\hat{y} + b\hat{x}$$

for any $a, b \in \mathbb{R}$ and $z \in \mathbb{R}^k$ with $z \perp x$ and $z \perp y$. Also let $\angle(x, y)$ denote the angle between vectors $x$ and $y$ (always between 0 and $\pi$).

It can be calculated that

$$Q_{x,y} = \frac{\pi - \angle(x,y)}{2\pi} I_k + \frac{\sin \angle(x,y)}{2\pi} M_{x,y}.$$

This is the expression we'll manipulate. It can be easily shown that $\angle(x, y)$ is Lipschitz, so the first term is Lipschitz. The remaining difficulty is then in showing that $(\sin\theta)M_{x,y}$ is Lipschitz. We start by proving this under some weak assumptions, i.e. essentially that $x, y, \tilde{y}$ are "in general position".

**Lemma D.1.** *Let $x, y, \tilde{y} \in S^{k-1}$. Let $\theta = \angle(x, y)$ and $\phi = \angle(x, \tilde{y})$. Assume that $\theta, \phi \notin \{0, \pi\}$. Then*

$$\|(\sin\theta)M_{x,y} - (\sin\phi)M_{x,\tilde{y}}\| \le \sqrt{79}\,\|y - \tilde{y}\|_2\,.$$

*Proof.* Fix an orthonormal basis for $\text{span}\{x, y, \tilde{y}\}$ in which

$$x = (1, 0, 0)$$

$$y = (\cos\theta, \sin\theta, 0)$$

$$\tilde{y} = (\cos\phi, \sin\phi\cos\alpha, \sin\phi\sin\alpha).$$

Let $d = \|y - \tilde{y}\|_2$ and observe that $\sin^2 \alpha \le d^2/\sin^2\phi$. Moreover if $\beta = \angle(y, \tilde{y})$ then $\beta \le 2\sqrt{2 - 2\cos\beta} = 2d$, so $|\theta - \phi| \le \beta \le 2d$. Since sin and cos are 1-Lipschitz it follows that $|\sin\theta - \sin\phi| \le 2d$ and $|\cos\theta - \cos\phi| \le 2d$.

Let $u \in S^{k-1}$. It can be checked that in this basis,

$$M_{x,y}u = (u_1\cos\theta + u_2\sin\theta, u_1\sin\theta - u_2\cos\theta, 0)$$

and

$$M_{x,\tilde{y}}u = (u_1 \cos\phi + u_2 \sin\phi \cos\alpha + u_3 \sin\phi \sin\alpha,$$
$$u_1 \sin\phi \cos\alpha - u_2 \cos\phi \cos^2\alpha - u_3 \cos\phi \sin\alpha \cos\alpha,$$
$$u_1 \sin\phi \sin\alpha - u_2 \cos\phi \sin\alpha \cos\alpha - u_3 \cos\phi \sin^2\alpha).$$

Indeed, we see that $M_{x,y}u \in \text{span}\{x,y\}$ and $(M_{x,y}u) \cdot x = u \cdot y$ and $(M_{x,y}u) \cdot y = u \cdot x$ (and similarly for $M_{x,\tilde{y}}u$). But now

$$((\sin\theta)M_{x,y}u - (\sin\phi)M_{x,\tilde{y}}u)_1^2$$
$$= (u_1(\sin\theta\cos\theta - \sin\phi\cos\phi) + u_2(\sin^2\theta - \sin^2\phi\cos\alpha) - u_3\sin^2\phi\sin\alpha)$$
$$\leq (\sin\theta\cos\theta - \sin\phi\cos\phi)^2 + (\sin^2\theta - \sin^2\phi\cos\alpha)^2 - \sin^4\phi\sin^2\alpha$$
$$\leq (|\sin\theta - \cos\theta| + |\sin\phi - \cos\phi|)^2 + (|\sin^2\theta - \sin^2\phi| + |\sin^2\phi(1 - \cos^2\alpha)|)^2$$
$$\quad + \sin^4\phi\sin^2\alpha$$
$$\leq 33d^2.$$

Similarly,

$$((\sin\theta)M_{x,y}u - (\sin\phi)M_{x,\tilde{y}}u)_2^2$$
$$\leq (\sin^2\theta - \sin^2\phi\cos\alpha)^2 + (\sin\theta\cos\theta - \sin\phi\cos\phi\cos^2\alpha)^2 + \sin^2\phi\cos^2\phi\sin^2\alpha\cos^2\alpha$$
$$\leq 16d^2 + (|\sin\theta\cos\theta - \sin\phi\cos\phi| + |\sin\phi\cos\phi\sin\alpha|)^2 + d^2$$
$$\leq 42d^2$$

and

$$((\sin\theta)M_{x,y}u - (\sin\phi)M_{x,\tilde{y}}u)_3^2$$
$$\leq \sin^4\phi\sin^2\alpha + \sin^2\phi\cos^2\phi\sin^4\alpha + \sin^2\phi\cos^2\phi\sin^2\alpha\cos^2\alpha$$
$$\leq 3d^2.$$

Therefore

$$\|(\sin\theta)M_{x,y}u - (\sin\phi)M_{x,\tilde{y}}u\|_2 \leq d\sqrt{79}.$$

Since this holds for all unit vectors $u$, the lemma follows. $\qquad\square$

It immediately follows that beyond relating $M_{x,y}$ to $M_{x,\tilde{y}}$ we can relate $M_{x,y}$ to $M_{\tilde{x},\tilde{y}}$.

**Corollary D.2.** *Let $x, y, \tilde{x}, \tilde{y} \in S^{k-1}$. Let $\theta = \angle(x,y)$ and $\tilde{\theta} = \angle(\tilde{x}, \tilde{y})$. Assume that $\theta, \tilde{\theta} \notin \{0, \pi\}$. Then*

$$\left\|(\sin\theta)M_{x,y} - (\sin\tilde{\theta})M_{x,\tilde{y}}\right\| \leq \sqrt{79}(\|x - \tilde{x}\|_2 + \|y - \tilde{y}\|_2).$$

*Proof.* Either $\angle(x, \tilde{y}) \neq 0$ or $\angle(y, \tilde{x}) \neq 0$, since otherwise $M_{x,y} = M_{\tilde{x},\tilde{y}}$. Without loss of generality suppose $\angle(x, \tilde{y}) \neq 0$. Then by the previous lemma, we can relate $M_{x,y}$ to $M_{x,\tilde{y}}$. And we can relate $M_{x,\tilde{y}}$ to $M_{\tilde{x},\tilde{y}}$. We get the claimed bound. $\qquad\square$

From this result the full claim is now easily derived.

**Lemma D.3.** *Let $x, y, \tilde{x}, \tilde{y} \in S^{k-1}$. Let $d = \max(\|x - \tilde{x}\|_2, \|y - \tilde{y}\|_2)$. Let $\theta = \angle(x,y)$ and $\tilde{\theta} = \angle(\tilde{x}, \tilde{y})$. Then*

$$\left\|(\sin\theta)M_{x,y} - (\sin\tilde{\theta})M_{\tilde{x},\tilde{y}}\right\| \leq 2\sqrt{79}d.$$

*Proof.* If $\sin\theta = \sin\tilde{\theta} = 0$ then the claim is clear. If both are nonzero it follows from the previous lemma. So now suppose without loss of generality that $\sin\tilde{\theta} = 0$ but $\sin\theta > 0$. We've shown in the previous lemma that $|\sin\theta - \sin\tilde{\theta}| \leq 4d$. Additionally, $\|M_{x,y}\| \leq 1$. Therefore

$$\left\|(\sin\theta)M_{x,y} - (\sin\tilde{\theta})M_{\tilde{x},\tilde{y}}\right\| = (\sin\theta)\|M_{x,y}\| \leq 4d.$$

This bound is sufficient. $\qquad\square$

**Lemma D.4.** *Let $x, y, \tilde{x}, \tilde{y} \in (3/2)\mathcal{B} \setminus (1/2)\mathcal{B}$. Let $d = \max(\|x - y\|, \|\tilde{x} - \tilde{y}\|)$. Then*

$$\|Q_{x,y} - Q_{\tilde{x},\tilde{y}}\| \leq O(\|x - \tilde{x}\|_2 + \|y - \tilde{y}\|_2).$$

*Proof.* Let $\hat{x} = x/\|x\|$ and $\hat{y} = y/\|y\|$. Similarly define $\hat{\tilde{x}}$ and $\hat{\tilde{y}}$. Note that normalizing at most doubles the distance. Therefore

$$\left\| (\sin\theta) M_{x,y} - (\sin\tilde{\theta}) M_{\tilde{x},\tilde{y}} \right\| = \left\| (\sin\theta) M_{\hat{x},\hat{y}} - (\sin\tilde{\theta}) M_{\hat{\tilde{x}},\hat{\tilde{y}}} \right\|$$
$$\leq 2\sqrt{79}(2d)$$

Additionally, let $\theta$ and $\tilde{\theta}$ be as defined previously. We have $|\theta - \tilde{\theta}| \leq 4(2d) = 8d$. Hence,

$$2\pi \|Q_{x,y} - Q_{\tilde{x},\tilde{y}}\| = \left\| (\tilde{\theta} - \theta) I_k + (\sin\theta) M_{x,y} - (\sin\tilde{\theta}) M_{\tilde{x},\tilde{y}} \right\|$$
$$\leq 8d + 4\sqrt{79}d.$$

Simplifying,

$$\|Q_{x,y} - Q_{\tilde{x},\tilde{y}}\| \leq 7d$$

as desired. $\qquad\square$

# E  Global landscape analysis

In this section, we explain why the Weight Distribution Condition arises, by sketching the basic theory of compressed sensing with generative priors. While our work applies to a number of different models in compressed sensing with generative priors (see Section 3.1 for details), we limit the exposition in this section to the *global landscape analysis* of the vanilla model, due to [8].

Let $G : \mathbb{R}^k \to \mathbb{R}^n$ be a fully connected ReLU neural network of the form

$$G(x) = \mathsf{ReLU}(W^{(d)}(\dots \mathsf{ReLU}(W^{(2)}(\mathsf{ReLU}(W^{(1)}x)))\dots)).$$

Let $x^* \in \mathbb{R}^k$ be an unknown latent vector. We wish to recover $x^*$ (or $G(x^*)$) from $m \ll n$ noisy linear measurements of $G(x^*)$. Specifically, for some measurement matrix $A \in \mathbb{R}^{m \times n}$ and noise vector $e \in \mathbb{R}^m$ we observe

$$y = AG(x^*) + e.$$

The scenario of interest is that the number of measurements $m$ is much less than the output dimension $n$, but is slightly more than the latent dimension $k$. Each $W^{(i)}$ is a matrix with dimension $n_i \times n_{i-1}$, such that $n_0 = k$ and $n_d = n$. The noise is arbitrary, and recovery bounds will depend on $\|e\|$.

The aim of [8] is to show that the empirical squared-loss

$$f(x) = \frac{1}{2} \|AG(x) - y\|_2^2$$

has no critical points except the true solution $x^*$, and a rescaled vector $-\rho_d x^*$. Thus, gradient descent would recover $x^*$ up to a global rescaling. This result is strengthened in subsequent work [10], but it contains the main ideas, which we outline now. See Section 2 of [8] for more details.

Ignoring issues of non-differentiability, the gradient is

$$\nabla f(x) = \left( \prod_{i=d}^{1} W_{+,x^{(i)}}^{(i)} \right)^T A^T A \left( \left( \prod_{i=d}^{1} W_{+,x^{(i)}}^{(i)} \right) x - \left( \prod_{i=d}^{1} W_{+,(x^*)^{(i)}}^{(i)} \right) x^* \right)$$

where $x^{(i)} = W^{(i-1)} \cdots W^{(1)} x$ and $(x^*)^{(i)} = W^{(i-1)} \cdots W^{(1)} x^*$. Next, if $A$ satisfies a certain restricted isometry condition, it follows that $A^T A$ is approximately the identity on the range of $G$, so

$$\nabla f(x) \approx \left( \prod_{i=d}^{1} W_{+,x^{(i)}}^{(i)} \right)^T \left( \left( \prod_{i=d}^{1} W_{+,x^{(i)}}^{(i)} \right) x - \left( \prod_{i=d}^{1} W_{+,(x^*)^{(i)}}^{(i)} \right) x^* \right).$$

Next, we need to show that this approximation concentrates around its mean. Consider the second term in the difference (the first is no more complicated):

$$(W_{+,x^{(1)}}^{(1)})^T \cdots (W_{+,x^{(d)}}^{(d)})^T W_{+,(x^*)^{(d)}}^{(d)} \cdots W_{+,(x^*)^{(1)}}^{(1)} x.$$

The proof that this product concentrates is by induction on $d$. Each step collapses the inner-most pair $(W_{+,x}^{(i)})^T W_{+,y}^{(i)}$ in the product, using the Weight Distribution Condition (which bounds $(W_{+,x}^{(i)})^T W_{+,y}^{(i)} - \mathbb{E}(W_{+,x}^{(i)})^T W_{+,y}^{(i)})$ to replace the pair with their expectation.

Finally, once the approximation for $\nabla f(x)$ has been further approximated by its (deterministic) expectation, the expectation is analyzed algebraically.

## F Extensions

### F.1 Gaussian noise

The CS-DGP problem (Compressed Sensing with a Deep Generative Prior) can be modified to the Gaussian noise setting, i.e. the noise vector $e \in \mathbb{R}^m$ has distribution $e \sim N(0, \sigma^2)^m$. In this setting it has been shown that there is an efficient algorithm estimating $x^*$ up to $\tilde{O}(\sigma^2 k/m)$ (ignoring logarithmic terms and dependence of the depth of the network), so long as the measurement matrix $A$ is random and the weight matrices satisfy the Weight Distribution Condition [9]. A corollary was that if the neural network was logarithmically expansive and had Gaussian random weights, then efficient recovery was possible. Our result directly yields an improvement, implying that constant expansion suffices.

### F.2 One-bit recovery

One-bit recovery with a neural network prior, introduced in [16], has the following formal statement. Let $G : \mathbb{R}^k \to \mathbb{R}^n$ be a neural network of the form

$$G(x) = \mathsf{ReLU}(W^{(d)}(\ldots \mathsf{ReLU}(W^{(2)}(\mathsf{ReLU}(W^{(1)}x)))\ldots)).$$

Let $x^* \in \mathbb{R}^k$ be an unknown latent vector. We wish to recover $x^*$ from $m$ one-bit noisy measurements of $G(x^*)$. Specifically, we observe a sign vector

$$y = \mathrm{sign}(AG(x^*) + \xi + \tau)$$

where $A \in \mathbb{R}^{m \times n}$ is a random measurement matrix, $\xi \in \mathbb{R}^m$ is a noise vector, and $\tau \in \mathbb{R}^k$ is a random quantization threshold. In this setting, global landscape analysis of the loss function can be performed, and it can be shown that if each weight matrix satisfies the WDC then there are no spurious critical points outside a neighborhood of $x^*$, a neighborhood of some negative scalar multiple of $x^*$, and a neighborhood of $0$ [16].

Once again, our result implies that the analysis can go through when each weight matrix has i.i.d. Gaussian entries and constant expansion in dimension (whereas previously logarithmic expansion was required).

### F.3 Phase retrieval

Phase retrieval with a neural network prior, introduced in [7], has the following formal statement. Let $G : \mathbb{R}^k \to \mathbb{R}^n$ be a neural network of the form

$$G(x) = \mathsf{ReLU}(W^{(d)}(\ldots \mathsf{ReLU}(W^{(2)}(\mathsf{ReLU}(W^{(1)}x)))\ldots)).$$

Let $x^* \in \mathbb{R}^k$ be an unknown latent vector. We wish to recover $x^*$ from $m$ phaseless noisy measurements of $G(x^*)$. Specifically, we observe

$$y = |AG(x^*)|$$

where $A \in \mathbb{R}^{m \times n}$ is a measurement matrix. As in the prior two examples, global landscape analysis can be performed if each weight matrix satisfies the WDC (and $A$ satisfies a certain isometry condition) [7]. Thus, our contribution extends the analysis to Gaussian matrices with constant expansion.

### F.4  Deconvolutional neural networks

It is shown in [14] that if $G$ is a two-layer deconvolutional neural network with Gaussian weights and logarithmic expansion in the number of channels, then (under certain other moderate conditions), the empirical risk function is well-behaved. This result again applies the WDC for Gaussian random matrices as a black box, so our result decreases the requisite expansion to a constant.