[Reviews · NeurIPS 2020]

Review 1

Summary and Contributions: This paper is about compressed sensing (CS) under generative priors. In such a problem, undersampled linear measurements of a signal of interest are provided, and the signal is sought. The mathematical ambiguity is resolved by finding the feasible point that is in the range of a trained generative model (such as a GAN), which is itself computed by solving an empirical risk minimization. Existing theory establishes a convergence guarantee of an efficient algorithm under an appropriate random model for the weights of the generative prior. The convergence guarantee assumes that the generative model is a multilayer perceptron where the width of each layer grows log-linearly. As real neural networks can succeed despite expanding at every layer, it is important to extend the existing theory to less restrictive assumptions regarding network expansivity. The present work establishes comparable convergence guarantees for CS with generative priors with only linear expansivity over the layers of a network. To do this, the authors introduce a novel technique for establishing concentration of random functions. The work of this paper immediately generalizes several existing works, including compressive sensing with generative priors and phase retrieval with generative priors. The authors additionally contribute a tight lower bound on the expansivity of one-layer generative models in order to allow invertibility (of generative models) without noise or compressive measurements.

Strengths: This paper presents an original and creative new proof technique for establishing concentration of random functions. Such concentration results are common in machine learning and signal processing, and this technique could potentially be extended to many new contexts. The paper broadens the applicability of rigorous recovery guarantees for deep generative neural networks when used as priors for inverse problems. These improvements immediately provide advances to multiple important inverse problems.

Weaknesses: Along the main theme of the paper, I do not see any significant weaknesses. My only relatively minor issue with the paper is regarding the lower bound on generative model expansivity. There is an apparent contradiction between the provided lower bound that expansivity must be at least a factor of 2 at each layer (Line 165 and Section C) and the comment on line 65 that "The expansivity condition is a major limitation of the prior theory, since in practice neural networks do not expand at every layer." The paper could be improved by commenting on / resolving this contradiction. In Section C, the authors state that "it suffices to consider a one-layer neural network." Under the presumption that the expansivity at each layer is constant, this comment is true, but it is misleading because expansivity can in generally vary between layers, as it does in real networks. If the first layer is expansive but the subsequent layers are not expansive, then there is not necessarily a problem. If two different activations of the second layer give identical activations in the third layer, it is possible that only one of these second-layer activations lies in the range of the earlier layers).

Correctness: Yes

Clarity: Yes

Relation to Prior Work: Yes, though I think that in Line 84, the authors should add a reference to "A Geometric Analysis of Phase Retrieval" by Sun, Qu, Wright, Foundations of Computational Mathematics, 2017. https://dx.doi.org/10.1007/s10208-017-9365-9

Reproducibility: Yes

Additional Feedback: Overall, great work. Unclear sentence: Line 79 "The Gaussian noise setting admits an efficient algorithm with recovery error Theta(k/m)" does not provide sufficient clarity of context. You may want to mention the denoising problem for increased clarity. Typos: Line 225 - "family of functions f_w(x) = w.x is (eps, c eps , 1/2)-pseudo-Lipschitz". A factor of two might be missing (or perhaps a different constant c is used with the previous line). Line 562 in Supplemental materials is missing a ")" ******** After Reading Author Responses I am satisfied with the author response, and I still believe the paper is a good submission that should be accepted to this conference.


Review 2

Summary and Contributions: This is in fact a paper on uniform concentration. The question is as follows. Given a family of functions {f_t}_{t in T} (where T is a probability space) over some space X, we know that f_t(x) is close to f(x) with high probability for a random t, and wish to show that with high probability, f_t(x) is uniformly close to f(x) simultaneously for all x in X. Usually one would assume that f_t and f are Lipschitz (over T x X and over X, respectively), and use a net argument. However, this fails to give the tight bound of the dimension of a Gaussian random matrix that satisfies the weight distribution condition (WDC). The looseness comes from the fact that treating f_t(x) as a function over T x X and requiring Lipschitzness would incur a large Lipschitz constant. The innovation in this paper is that, while it still resorts to an eps-net type argument, it does not use the same standard ball as the neighbourhood for the net, and does not use the same net for all f_t. Instead, for each f_t, it uses potentially a different shape of neighbourhood to form the net and randomizes the net points. This sort of decouples the net from the parameter t, allowing for a tighter bound.

Strengths: This technique is potentially useful for many other problems which relies on uniform concentration. Various communities may be benefited from this technique.

Weaknesses: Nothing really.

Correctness: I read the main body of the paper and found the approach convincing and so I believe it is correct. I did not read the proofs in the supplementary material.

Clarity: The paper is also well-written. I find it very pleasant to read. Small typos: Line 101: “uniformly near” sounds incomplete. Maybe “uniformly concentrated near”? Line 255: u^T G(x,y) u^T should be u^T G(x,y) u Line 258: both bounds (1) and (2) refer to the two bounds listed above, instead of equations (1) and (2), right? Maybe use (a) and (b) here to avoid confusion

Relation to Prior Work: The difference in approach is clearly stated.

Reproducibility: Yes

Additional Feedback:


Review 3

Summary and Contributions: This paper considers the problem of retrieving a signal that can be expressed as the output of a neural network with random weights applied to a low-dimensional vector. Previous work developed guarantees for this problem under the condition that the dimension of the weight matrices in the network grow by a logarithmic factor. Here the authors show that it is sufficient for the dimension to grow by a constant factor. To this end they introduce a concept called pseudo-Lipschitzness and use it to prove a concentration bound.

Strengths: The paper is written beautifully. The authors explain their contributions extremely clearly. The mathematical content is quite interesting and well explained. The concentration bound may be useful to analyze other problems.

Weaknesses: The signal model considered in the paper is essentially of theoretical interest: the signal is the output of a neural network with random weights, encoded by a low-dimensional vector at the input. Its connection to signals or measurement models of practical interest is unclear, which will limit the impact of the paper and its potential audience to very mathematically-oriented readers. The mathematical contribution may be interesting in itself but its applications seem somewhat restricted in scope.

Correctness: Yes, as far as I can tell.

Clarity: Yes, extremely well written.

Relation to Prior Work: Yes.

Reproducibility: Yes

Additional Feedback: I have updated my score after reading the author feedback.

[Author Response · NeurIPS 2020]

*We thank all reviewers for their comments. We apologize for the typos; all these minor points will be dealt with in the revised version of our article.*

R#2: My only relatively minor issue with the paper...range of the earlier layers)..

Indeed, we did not clearly explain what exactly the lower bound proves, in two different ways:

– As you point out, we incorrectly asserted that it suffices to consider a one-layer neural network. Our lower bound does only apply to one-layer neural networks. If there are many layers, some expansive and some not, then things might still be fine; we do not know and it is indeed an interesting open problem.

– Our lower bound is only against recovering the latent parameter $x^*$, not against recovering the image $G(x^*)$. For recovering the image, it's not clear (to us, at least) that any expansion should be necessary – it's not information-theoretically necessary, and although global landscape analysis probably fails without expansion, there may be other algorithms.

Thank you for this interesting comment, we will clarify/fix these issues in the final version of our paper.

I think that in Line 84, the authors should add a reference to "A Geometric Analysis of Phase Retrieval". We believe that we have a reference to this exact paper on line 84 :-).

R#3: We really appreciate your positive feedback!

R#4: The signal model considered in the paper is essentially of theoretical interest: the signal is the output of a neural network with random weights, encoded by a low-dimensional vector at the input. Its connection to signals or measurement models of practical interest is unclear, which will limit the impact of the paper and its potential audience to very mathematically-oriented readers. The mathematical contribution may be interesting in itself but its applications seem somewhat restricted in scope.

The generative prior model (i.e. "signal is output of neural network with low latent dimension") is in fact of significant practical interest. It has been intensively studied in recent years in the context of compressed sensing, inpainting, and other image recovery problems [2]. Indeed, much empirical evidence suggests that the generative prior can enable image recovery with far fewer samples than sparsity-based priors (e.g. in the wavelet basis).

Our goal in this paper, and many papers before us [4, 3, 5], is to theoretically analyze the already established practical success of the generative prior model. Our contribution is to strengthen the existing results by making weaker assumptions. This brings us closer to rigorously explaining the practical success of this method.

Why Gaussian random weights? First, it's a common theoretical assumption, and success at analyzing Gaussians motivates study of more realistic distributions. Second, some works have shown that even trained neural networks have weights which look Gaussian in important ways (e.g. singular values of the weight matrices) [1]. Third, random initialization is a technique commonly used in practice; understanding the theoretical properties of a network at initialization is necessary to understand the properties after training. Indeed many theoretical works on overparametrized networks argue that the weights don't move a lot after training. Finally, when a trained GAN is unavailable, a practical approach, called Deep Image Prior is to take a network with randomly assigned weights, and use that as a regularizer. This is our setting. For further motivation for Gaussian weights, we refer to prior work in this area.

# References

[1] Sanjeev Arora, Yingyu Liang, and Tengyu Ma. Why are deep nets reversible: A simple theory, with implications for training. *arXiv preprint arXiv:1511.05653*, 2015.

[2] Ashish Bora, Ajil Jalal, Eric Price, and Alexandros G Dimakis. Compressed sensing using generative models. In *Proceedings of the 34th International Conference on Machine Learning-Volume 70*, pages 537–546. JMLR. org, 2017.

[3] Paul Hand, Oscar Leong, and Vlad Voroninski. Phase retrieval under a generative prior. In *Advances in Neural Information Processing Systems*, pages 9136–9146, 2018.

[4] Paul Hand and Vladislav Voroninski. Global guarantees for enforcing deep generative priors by empirical risk. *IEEE Transactions on Information Theory*, 66(1):401–418, 2019.

[5] Shuang Qiu, Xiaohan Wei, and Zhuoran Yang. Robust one-bit recovery via relu generative networks: Improved statistical rates and global landscape analysis. *arXiv preprint arXiv:1908.05368*, 2019.


[Meta-Review · NeurIPS 2020]

In compressed sensing with a random multilayer ReLU neural network as prior, this paper shows that constant expansivity of the weight matrices of the neural network, as opposed to the "strong" expansivity (i.e., with a logarithmic factor) in existing studies, suffices for the existence of a gradient-descent based algorithm with a theoretical recovery guarantee (Theorem 1.1). To prove it, this paper introduced and utilized the novel notion of pseudo-Lipschitzness (Definition 4.2). This paper furthermore succeeded in obtaining several generalizations of Theorem 1.1, as stated informally in Theorem 1.2. The three reviewers rated this paper well above the acceptance threshold. They also agreed that the proof technique developed in this paper will have wider applicability, as well as that this paper is very clearly written. Two of them acknowledged that there is no significant weakness. I am therefore glad to recommend acceptance of this paper.